

# Extracting automata from neural networks using active learning

Zhiwu Xu[1], Cheng Wen[1], Shengchao Qin[1,2] and Mengda He[2]

[1] College of Computer Science and Software Engineering, Shenzhen University, Shenzhen, China
[2] School of Computing, Engineering and Digital Technologies, Teesside University, Middlesbrough, United Kingdom

## ABSTRACT

Deep learning is one of the most advanced forms of machine learning. Most modern deep learning models are based on an artificial neural network, and benchmarking studies reveal that neural networks have produced results comparable to and in some cases superior to human experts. However, the generated neural networks are typically regarded as incomprehensible black-box models, which not only limits their applications, but also hinders testing and verifying. In this paper, we present an active learning framework to extract automata from neural network classifiers, which can help users to understand the classifiers. In more detail, we use Angluin's $L^*$ algorithm as a learner and the neural network under learning as an oracle, employing abstraction interpretation of the neural network for answering membership and equivalence queries. Our abstraction consists of value, symbol and word abstractions. The factors that may affect the abstraction are also discussed in the paper. We have implemented our approach in a prototype. To evaluate it, we have performed the prototype on a MNIST classifier and have identified that the abstraction with interval number 2 and block size $1 \times 28$ offers the best performance in terms of $F1$ score. We also have compared our extracted DFA against the DFAs learned via the passive learning algorithms provided in LearnLib and the experimental results show that our DFA gives a better performance on the MNIST dataset.

# INTRODUCTION

Deep learning is one of the most advanced forms of machine learning, which has been applied to various fields, including computer vision, speech recognition, natural language processing, audio recognition, social network filtering, machine translation, bioinformatics, drug design and board game programs (*Schmidhuber, 2014*; *Lecun, Bengio & Hinton, 2015*). Most modern deep learning models are based on an artificial neural network, such as deep neural networks (DNN), deep belief networks, convolutional neural networks (CNN) and recurrent neural networks (RNN). Benchmarking studies reveal that neural networks have produced results comparable to and in some cases superior to human experts.

However, the generated neural networks are typically regarded as incomprehensible black-box models. They are in practice unlikely to generalise exactly to the concept being trained, and what they eventually learn actually is unclear (*Omlin & Giles, 2000*). The opaqueness of neural networks not only limits their applications, but also hinders

Corresponding author
Shengchao Qin,
shengchao.qin@gmail.com

testing and verifying. Indeed, several lines of work attempt to glimpse into the black-box networks, especially RNN (*Zeng, Goodman & Smyth, 1993*; *Sanfeliu & Alquezar, 1994*; *Tiňo & Šajda, 1995*; *Omlin & Giles, 1996*; *Frasconi et al., 1996*; *Gori et al., 1998*; *Cechin, Simon & Stertz, 2003*; *Cohen et al., 2017*). They induce rules that mimic the blackbox neural networks as closely as possible, by exploring the possible state vectors of networks, which is often practically impossible at present.

Active learning (*Angluin, 1987*) can learn finite automata (sets of words) from a *minimally adequate teacher* (MAT), an oracle capable of answering the so-called *membership* and *equivalence* queries, which has been successfully applied to numerous practical cases in different domains (*Vaandrager, 2017*; *Aichernig et al., 2018*). Recently, *Weiss, Goldberg & Yahav (2018)* adopted active learning to extract automata from neural networks. But the network systems under learning are RNN acceptors on small regular languages: an input at a time is considered as a symbol, and thus a sequence of inputs is a word. Hence, their approach is not suitable to other neural networks in practice such as CNN, since not all neural networks perform on discrete-time symbolic data.

In this paper, we present an active learning framework to extract automata from neural network classifiers, which is inspired by *Weiss, Goldberg & Yahav (2018)'s* work. But different from their work, we consider each input as a word (i.e., a sequence of symbols) rather than a symbol, via abstraction interpretation, which consists of value, symbol and word abstractions. Therefore, the system under learning (SUL) here can be any neural network. Indeed, we assume that we have no idea about the framework of neural networks: we can not access the state-vectors nor we do not know the relations between two consecutive inputs. For simplicity, we focus on network-acceptors, that is, binary neural network classifiers, since multi-class classifiers can be reduced into several binary classifiers[1].

Abstraction is the key for scaling model learning methods to realistic application (*Vaandrager, 2017*), so the key idea of our approach is to define an abstraction for the neural network classifier under learning. Our abstraction consists of three layers: (1) value abstraction: each value in an input array is mapped into an integer via partitioning; (2) symbol abstraction: a block of multi-dimensional integer array is abstracted as a symbol; and (3) word abstraction: the whole input array is encoded into a word. We also discuss the factors that may affect the abstraction.

Next, we present how to instantiate the active learning framework on neural networks, in particular the membership and equivalence queries (*Vaandrager, 2017*). Membership queries can be answered by the neural networks via the word concretization function: we concretise the word that is being queried and then feed the concretised data into the neural network under learning. Equivalence query is more of a challenge, because there is no finite interpretation for neural networks (*Weiss, Goldberg & Yahav, 2018*). To address this, we use as an abstract model the automaton that is learned passively from some test data in the training dataset and then perform the equivalence query against the abstract model. If no words that separate the hypothesis and the abstract model are found, then the answer for the equivalence query is *yes*. Note that, when a counterexample

---

[1] This may raise a comparative performance problem, which is not the scope of this paper.

is found, it may be not that the hypothesis is incorrect, but rather that the abstract model is not precise enough and needs to be refined.

Finally, we have implemented our approach in Java, wherein we use the library *LearnLib* (*Howar et al., 2012*) to implement the active learning framework. To evaluate our approach, we conducted a series of experiments on a classifier for the MNIST dataset, a large database of handwritten digits that is commonly used for training various image processing systems. We first test the measures of the MNIST classifier, namely, safety, conflict, the size of alphabet and the length of words, under the abstractions with different interval numbers (i.e., the number of partitioning) and block sizes, and have identified some suitable abstractions. Secondly, we conduct some experiments to learn DFAs from the MNIST classifier with the suggested abstractions. The results shows that the abstraction with interval number 2 and block size $1 \times 28$ offers the best performance in terms of $F1$ score. At last, we also conduct the experiments to compare our resulted DFA against the DFAs learned via the passive learning algorithms (see "Passive Learning") provided in LearnLib and the MNIST classifier itself. Although worse than the classifier, our DFA gives a better performance than the other DFAs in our experiments. Nevertheless, there are still some limitations for our approach.

In summary, our contributions are as follows:

- We have proposed an MAT framework to extract automata from neural networks, employing abstraction interpretation of the neural network for answering membership and equivalence queries.
- We have conducted several experiments on a MNIST classifier, and the experimental results show that our approach is viable, and the resulted DFA has a better performance in terms of F1 score than the DFAs learned via the passive learning algorithms provided in LearnLib on the MNIST dataset.

The remainder of this paper is organised as follows. "Preliminary" gives the preliminaries of DFA and active learning. "Approach" describes our approach, followed by the experimental results in "Experiments". "Limitations" discusses some limitations of our approach. "Related Work" presents the related work, followed by some concluding remarks in "Conclusion".

## PRELIMINARY

In this section, we present the notion about neural networks, deterministic finite automata, active learning and passive learning.

### Neural networks as functions

Neural network models can be viewed as mathematical models defining a possible non-linear function $\mathcal{N} : X \rightarrow Y$, where $X$ is the input and $Y$ is the output. In more detail, $\mathcal{N}$ can be defined as a composition of other (layer) functions, which can further be decomposed into other functions. This can be conveniently represented as a network structure, with arrows depicting the dependencies between functions. In this paper we assume that the framework of neural networks is unknown and only network-acceptors

are considered. So the function we consider here is the one representing the whole network $\mathcal{N} : X \rightarrow Bool$, where $X$ is a multi-dimensional array. We say an input data $X$ is *positive* if $\mathcal{N}(X) = true$, and otherwise *negative*.

## Deterministic finite automata

### Definition 2.1

*A deterministic finite automaton (DFA) is a 5-tuple (Q, σ, δ, $q_0$, F), where*

- *Q is a finite set of states,*
- *σ is a finite set of input symbols and is called the alphabet,*
- *δ : Q × Σ → Q is the transition function,*
- *$q_0$ ∈ Q is the starting state,*
- *F ⊆ Q is the set of accepting states.*

A *word* or *string* over an alphabet σ is a finite sequence of symbols from σ. The *length* of a word is the number of symbols it contains. Note that a word can be empty: the empty word, denoted as $\varepsilon$, has length 0 and contains no symbols.

### Definition 2.2

*Let $\mathcal{M} = (Q, \Sigma, \delta, q_0, F)$ be a DFA and $w = a_1 a_2 \ldots a_n$ be a word of length n over σ. The automaton $\mathcal{M}$ accepts the word w if and only if there exists a sequence of states $r_0, r_1, \ldots, r_n$ with the following conditions:*

- *$r_0 = q_0$*
- *$r_{i+1} = \delta(r_i, a_{i+1})$, for i = 0, …, n − 1*
- *$r_n \in F$.*

The set of words recognised by a DFA $\mathcal{M}$, called the language of $\mathcal{M}$, is the following set:

$$L(\mathcal{M}) = \{w \in \Sigma^* | w \text{ is accepted by } \mathcal{M}\}$$

## Active learning framework

*Angluin (1987)* proposed the first active learning algorithm, the $L^\star$ algorithm, to learn finite automata from a MAT in 1987, and today all the most efficient learning algorithms that are being used follow Angluin's approach. In the following, we briefly introduce the MAT framework.

Finite automata can be learned precisely from a MAT, that is, an oracle capable of answering the so-called *membership* and equivalence queries:

- membership queries: the learner asks whether a given word is accepted by the automaton or not, and the teacher answers with the result.
- equivalence queries: the learner asks whether a given hypothesis automaton $\mathcal{H}$ is equal to the automaton model $\mathcal{M}$ held by the teacher. The teacher answers yes if this is the case. Otherwise she answers no and supplies a word, the so-called *counterexample*, on which the hypothesis automaton $\mathcal{H}$ and the automaton model $\mathcal{M}$ disagree.

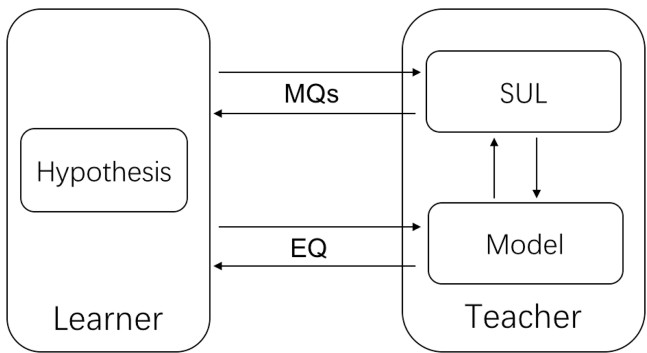

**Figure 1  The MAT framework.**     

The MAT framework is shown in Fig. 1. Initially, the learner knows the static interface of the SUL, that is, the sets of input (i.e., multi-dimensional array for neural networks) and output (i.e., *yes* or *no* for recognizers). Then the learner starts to ask a sequence of membership queries (MQs) and receives the corresponding responses from the teacher. After a "sufficient" number of queries, the learner builds a hypothesis $\mathcal{H}$ from the obtained information, and then sends an equivalence query (EQ). If the teacher answers yes, then the hypothesis $\mathcal{H}$ is returned. Otherwise, the learner refines the information with the returned counterexample, and continues on querying.

## Passive learning

Different from active learning, passive learning constructs automata from sets of examples directly. Many approaches in grammatical inference can be described as passive learning. In the paper, we consider the polynomial-time RPNI algorithms provided in the library *LearnLib* (*Howar et al., 2012*).

*Oncina & Garca (1992)* proposed the Regular Positive and Negative Inference (RPNI) algorithm for DFA learning. RPNI starts with a prefix tree acceptor, a tree-like DFA built from the learning examples by taking all the prefixes of the examples as states, and then greedily creates clusters of states (by merging) in order to come up with an automaton that is always consistent with the examples. Two heuristic strategies can be employed in state merging: Evidence Driven State Merging (EDSM) (*Cicchello & Kremer, 2002*) and Minimum Description Length (MDL) (*Adriaans & Jacobs, 2006*).

## APPROACH

In this section, we present an active learning framework to extract automata from neural network classifiers. Our framework is shown in Fig. 2, which is a classic MAT framework with an abstraction[2]. In a nutshell, we make an abstraction between the learner and the SUL. When the membership queries are sent, the abstraction maps the abstract words into the concrete ones (i.e., the up arrow in Fig. 2), which are then fed into the neural network under learning. When the equivalence queries are sent, the abstraction does the opposite (i.e., the down arrow in Fig. 2) and from the abstract words an abstract representation is built for checking. In the following, we explain how to define an abstraction for neural networks and how to instantiate the active learning framework on neural networks.

[2] Some papers use the term *mapper* (*Vaandrager, 2017*).
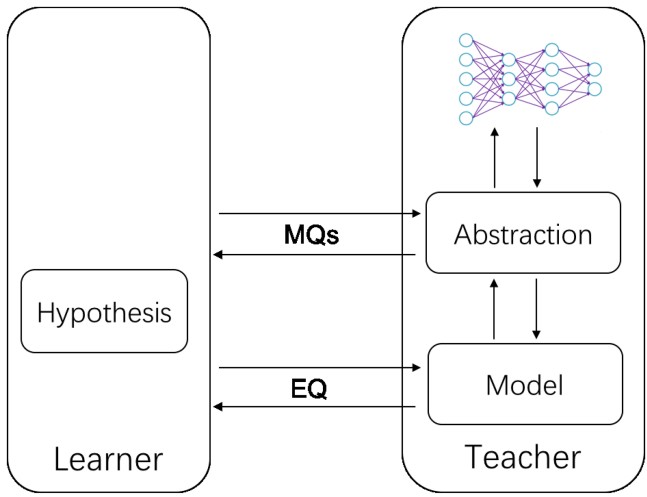

**Figure 2 Framework of our approach.**

## Abstraction

For simplicity, we focus on network-acceptors, that is, there are only two outputs for the SUL, and thus we do not need to abstract them. In other words, only the inputs need to be abstracted. Generally, the inputs of neural network classifiers are always multi-dimensional arrays. As mentioned in "Introduction", we aim to abstract an input as a word, rather than a symbol. So the aim of the abstraction is to convert multi-dimensional arrays into words and vice-versa.

Just like the serialization of multi-dimensional arrays, a naive and simple solution to the abstraction is to convert the input multi-dimensional array into a 1-dimensional array in row (or column) major order, and then concatenate the string representation of each value in the converted array in order, yielding a word. For example, Fig. 3 shows an array with size $4 \times 4$, on which the simple abstraction is applied. However, there are two problems for this solution: (1) the size of alphabet may be too large, even infinite. (2) the length of the abstracted word may be too long. Both of them can make the automata learning too time-consuming.

So for practicality, we propose a three-layer abstraction, which consists of:

- value abstraction: each value in an input array is mapped into an integer via partitioning, which helps reduce the size of alphabet;
- symbol abstraction: a block of multi-dimensional integer array is abstracted as a symbol, which enables us to reduce the length of word;
- word abstraction: the whole input array is encoded into a word, wherein value abstraction and symbol abstraction are applied.

### *Value abstraction*

In order to reduce the size of alphabet, inspired by *Omlin & Giles (1996)* work, we first split the values of the input space into *n* (equal) intervals, and map each interval into an integer, that is, the index of the corresponding intervals. Formally, let the input space be $\mathscr{I}$ and

| 0.7 | 0.4 | 0.6 | 0.8 |
| 0.3 | 0.5 | 0.7 | 0.9 |
| 0.6 | 0.8 | 0.2 | 0.4 |
| 0.7 | 0.1 | 0.3 | 0.5 |

$\longrightarrow$

| 0.7 | 0.4 | ⋯ | 0.3 | 0.5 |

$\downarrow$

$0.7, 0.4, ..., 0.3, 0.5$

**Figure 3  Example of simple abstraction.**

| 0.7 | 0.4 | 0.6 | 0.8 |
| 0.3 | 0.5 | 0.7 | 0.9 |
| 0.6 | 0.8 | 0.2 | 0.4 |
| 0.7 | 0.1 | 0.3 | 0.5 |

$\xrightarrow{\alpha_v}$

| 1 | 0 | 1 | 1 |
| 0 | 1 | 1 | 1 |
| 1 | 1 | 0 | 0 |
| 1 | 0 | 0 | 1 |

**Figure 4  Example of value abstraction.**

it be split into $n$ intervals $I_0, \ldots, I_{n-1}$. Then a value abstraction function $\alpha_v : \mathscr{I} \to \{0, \ldots, n-1\}$ is defined as follows:

$$\forall d \in \mathscr{I} . \alpha_v(d) = i \text{ such that } d \in I_i$$

This value abstraction function maps a concrete value in the input to an abstract integer. Figure 4 shows an example of value abstraction applying on the array given in Fig. 3, where the input space $\mathscr{I}$ is [0,1] and $\mathscr{I}$ is split into $I_0 = [0,0.5)$ and $I_1 = [0.5,1]$.

While for concretization, an abstract integer is mapped to its corresponding interval, that is, the value concretization function $\gamma_v : \{0, \ldots, n-1\} \to 2^{\mathscr{I}}$ is defined:

$$\gamma_v(i) = I_i$$

Both $\alpha_v$ and $\beta_v$ can be extended on sets of elements in a natural way:

$$f_v(S) = \bigcup_{d \in S} f_v(d)$$

where $f_v$ is $\alpha_v$ or $\beta_v$ .

It is easy to get that $\alpha_v(\gamma_v(i)) = i$ for each integer $i$, and $d \in \gamma_v(\alpha_v(d))$ for any given value $d$. Therefore, $(\alpha_v, \gamma_v)$ forms a Galois connection (*Nielson, Nielson & Hankin, 1999*). While in practice, especially when we query a word, we are unable to test all the values in the interval for each integer. For that, we randomly select at most $k_v$ (which can be dependent on the intervals) values to represent the corresponding interval. That is to say, we define a weak value concretization function:

$$\gamma_v^{\text{prac}}(i) = \{d_j \,|\, d_j \in I_i \text{ and } 0 \leq j < k_v\}$$

Obviously, the larger $k_v$ is, the closer $\gamma_v^{prac}(i)$ is to $\gamma_v(i)$ . So concerning Galois connections, the larger $k_v$, the better.

| 0.7 | 0.4 | 0.6 | 0.8 |
|---|---|---|---|
| 0.3 | 0.5 | 0.7 | 0.9 |
| 0.6 | 0.8 | 0.2 | 0.4 |
| 0.7 | 0.1 | 0.3 | 0.5 |

$\xrightarrow{vm^4}$

| 0.7 | 0.3 | 0.6 | 0.8 |
|---|---|---|---|
| 0.3 | 0.5 | 0.7 | 0.8 |
| 0.5 | 0.8 | 0.2 | 0.4 |
| 0.7 | 0.1 | 0.4 | 0.5 |

**Figure 5 Example of *k*-abstraction manipulation.**

But only Galois connections are not enough here. We also need to consider the *safety* of neural networks (*Huang et al., 2017*), that is, a vibration of values should not flap the outputs, since different values may be abstracted into an identity integer. In fact, the composition of the functions $\alpha_v$ and $\gamma_v$ can be viewed as a kind of manipulations (*Huang et al., 2017*). We say a *k*-value manipulation $vm^k$ with respect to $\alpha_v$ and $\gamma_v$ is a function such that for any input array *in*

$$vm^k(in) = in \cdot [d_i \mapsto d'_i]_{i \in \{1,\dots,k\}}$$

where $d_i \in in$ and $d'_i \in \gamma_v(\alpha_v(d_i))$. Intuitively, *k*-value manipulation replaces (at most) *k* values of the input array by some values, which share the same intervals with the corresponding original values. Figure 5 shows an example of 4-value manipulation applying on the input array in Fig. 4, where the input space $\mathscr{I}$ is [0,1] and it is split into $I_0 =$ [0,0.5) and $I_1 =$ [0.5,1]. And we say a network $\mathscr{N}$ is safe with respect to this value manipulation $vm^k$ if for every input array *in*

$$\mathscr{N}(vm^k(in)) = \mathscr{N}(in)$$

That is to say, performing this value manipulation should not result in a different classification. This requires that every interval $I_i$ should be as small as possible or the number *n* of intervals should be as large as possible. However, the safety cannot easily be preserved in practice, unless the abstraction is an identity function or the network is robust enough. So instead, we use a weak notation called σ-*safety*: we say a network $\mathscr{N}$ is σ-safe with respect to *k*-value manipulation $vm^k$ under a given input set *D* if

$$\frac{|\{in \in D \mid \mathscr{N}(vm^k(in)) \neq \mathscr{N}(in)\}|}{|D|} \leq \sigma$$

where $|D| \geq 1$.

### Symbol abstraction

After the value abstraction, each integer can be used as a symbol. But this could yield words that are too long to learn the model. So for scalability, we add a symbol abstraction, which abstracts input arrays into symbols by blocks. For simplicity, in this paper we consider 2-dimensional array with size *iRow* × *iCol*. We say a slice of an input array starting from the index $(ri, ci)$ to the index $(ri + oRow - 1, ci + oCol - 1)$ is a block, and the size of the block is *oRow* × *oCol*.

**Figure 6 Example of base-$n$ symbol abstraction.**

A natural way to abstract blocks into symbols is to map the blocks into one dimension in row (or column) major order and then encode the one dimension into a base-$n$ number (or a string consisting of the integers in the one dimension). We denote this mapping as $\alpha_s^B$. Figure 6 gives an example of $\alpha_s^B$ that are performed on the array that are obtained by the value abstraction shown in Fig. 4, where the number $n$ of intervals is 2 and the size of blocks is $2 \times 2$. Moreover, by decoding the base-$n$ number (or the string), it is easy to obtain the inverse mapping $\gamma_s^B$. It is clear $(\alpha_s^B, \gamma_s^B)$ forms a Galois connection. But a drawback of this solution is that the size of alphabet is $n^{oRow \times oCol}$, which could be too large in practice. For example, the size of alphabet of the example shown in Figure 6 is 16.

In this paper we use an alternative way to represent a block as its sum. In more detail, we define a symbol abstraction function $\alpha_s^S$ that maps integer blocks of size $oRow \times oCol$ into the sum of the integers in blocks:

$$\alpha_s^S(b) = \sum_{i \in b} i$$

It is easy to compute the set of the possible sums of blocks, that is, $\{0, \ldots, oRow \times oCol \times (n - 1)\}$. So the size of alphabet is $oRow \times oCol \times (n - 1) + 1$. Compared to the natural solution, the size of alphabet is quite smaller (from $n^{oRow \times oCol}$ reduced to $oRow \times oCol \times (n - 1) + 1$). Take the input array in Fig. 6 for example. Under this sum abstraction, its abstraction is shown in Fig. 7 and the size of alphabet is 5.

But it is pity that this mapping is not bijective. So in order to form Galois connections, similar to value abstraction, we define the inverse mapping from symbols (i.e. sums) to sets consisting of blocks of size $oRow \times oCol$ whose sum is exactly the symbol:

$$\gamma_s^S(\text{sum}) = \{b_j \mid \sum_{i \in b_j} i = \text{sum}\}$$

Likewise, these two functions can be lifted to sets of elements in a natural way. It is easy to get that $\alpha_s(\gamma_s(\text{sum})) = \text{sum}$ for each symbol $\text{sum}$, and $b \in \gamma_s(\alpha_s(b))$ for any given block $b$ of size $oRow \times oCol$. Therefore, $(\alpha_s, \gamma_s)$ forms a Galois connection. While for practicality, similar to value abstraction $\gamma_v$, we use a weak sum concretization function:

$$\gamma_s^{\text{Sprac}}(\text{sum}) = \{b_j \mid \sum_{i \in b_j} i = \text{sum and } 0 \leq j < k_s\}$$

**Figure 7 Example of sum symbol abstraction.**

That is, we select at most $k_s$ (which can be dependent on the block size and $n$) blocks to represent the corresponding sum. Clearly, the larger $k_s$ is, the closer $\gamma_s^{Sprac}(\text{sum})$ is to $\gamma_s^S(\text{sum})$. So concerning Galois connections, the larger $k_s$, the better.

In addition, the composition of $\alpha_s^S$ and $\gamma_s^S$ forms a manipulation as well, and the network $\mathcal{N}$ should be safe with respect to this manipulation. Formally, we say a $k$-shift manipulation $sm^k$ with respect to $\alpha_s^S$ and $\gamma_s^S$ is a function such that for any input array $in$

$$sm^k(in) = in \cdot [b_i \mapsto b_i']_{i \in \{1,\dots,k\}}$$

where $b_i$ is a block of size $oRow \times oCol$ belonging to $in$ and $b_i' \in \gamma_s^S(\alpha_s^S(b_i))$. And a network $\mathcal{N}$ is safe with respect to $sm^k$ if for every input array $in$

$$\mathcal{N}(sm^k(in)) = \mathcal{N}(in)$$

In a word, $k$-shift manipulation replaces (at most) $k$ blocks of the input array by some blocks, which share the same sums with the corresponding original blocks; and performing this $k$-shift manipulation should not result in a different classification. Similar to value manipulation, the safety cannot easily be preserved in practice. So we use a weak notation called $\sigma$-*safety*: a network $\mathcal{N}$ is $\sigma$-safe with respect to $sm^k$ under a given input set $D$ if

$$\frac{|\{in \in D | \mathcal{N}(sm^k(in)) \neq \mathcal{N}(in)\}|}{|D|} \leq \sigma$$

where $|D| \geq 1$. Considering the safety, the distance between two blocks of the same sum or the size of block should be as small as possible. Figure 8 shows an example of 1-shift manipulation applying on the input array in Fig. 7, where $n$ is 2 and the size of blocks is 2 × 2.

Finally, let us consider the alphabet size. As discussed above, the alphabet size for the abstraction function $\alpha_s^S$ ($\alpha_s$ resp.) is linear (exponential resp.) in the block size and the number of intervals $n$. So concerning the alphabet size, the smaller the block size and the interval number $n$, the better.

### Word abstraction

Finally, we split the input array into blocks, and map them into a sequence of symbols (i.e., a word) in row (or column) major order. Algorithm 1 shows the detail of the word abstraction function $\alpha_w$. The algorithm first invokes the value abstraction $\alpha_v$ to map the

**Figure 8 Example of *k*-shift manipulation.**

---

**Algorithm 1 Word abstraction function $\alpha_w(in)$.**

**Input:** an input *in*

**Output:** a word *w*

1: $in_I = \alpha_v(in)$

2: $w = \varepsilon$, $ri = 0$

3: **while** $ri + oRow < iRow$ **do**

4:     $ci = 0$

5:     **while** $ci + oCol < iCol$ **do**

6:         $w = w + \alpha_s^S(in_I[ri : ri + oRow][ci : ci + oCol])$

7:         $ci = ci + oCol$

8:     **end while**

9:     $ri = ri + oRow$

10: **end while**

11: **return** *w*

---

values in the input array into integers (Line 1). Then it slides over the integer array block by block (Lines 2–10) and maps each block into a symbol by the symbol abstraction $\alpha_s^S$ (Line 6). Note that here we use a narrow slide on the input array, that is, the blocks to be abstracted are fully contained in the input array. One can use the wide slide with zero-padding as well. Just like the convolution operation of CNN, one can further set the stride sizes for each dimension. In addition, one can further encode the sequence of symbols into a final word in a more compact format, such as run-length encoding (RLE).

As mentioned above, the symbol abstraction aims to reduce the length of words. According to the word abstraction, we have that the larger the block size, the shorter the word.

Figure 9 shows our abstraction applying on the input array given in Fig. 3, where the number *n* of intervals is 2 and the size of blocks is 2 × 2. Compared to the simple abstraction shown in Fig. 3, our abstraction yields words with smaller size of alphabet and shorter length.

The word concretization function $\gamma_w$, which is shown in Algorithm 2, does the opposite: it maps a sequence of symbols (i.e., a word) into a sequence of sets of blocks (Lines 5–7), and combines them into a set of arrays (Lines 8–10). Note that we require the length of

| 0.7 | 0.4 | 0.6 | 0.8 |
|---|---|---|---|
| 0.3 | 0.5 | 0.7 | 0.9 |
| 0.6 | 0.8 | 0.2 | 0.4 |
| 0.7 | 0.1 | 0.3 | 0.5 |

$\alpha_v$ $\dashrightarrow$

| 1 | 0 | 1 | 1 |
|---|---|---|---|
| 0 | 1 | 1 | 1 |
| 1 | 1 | 0 | 0 |
| 1 | 0 | 0 | 1 |

$\alpha_w \downarrow$          $\alpha_s^S \downarrow$

$2, 4, 3, 1$       encoding $\dashleftarrow$

| 2 | 4 |
|---|---|
| 3 | 1 |

**Figure 9  Example of word abstraction.**           

---

**Algorithm 2  Word Concretization Function γ(w).**

**Input:** a word $w$

**Output:** a set *matrix_set* of arrays

1: *rnum = iRow/oRow, cnum = iCol/oCol*

2: **if** *w.length* ≠ *rnum* × *cnum* **then**

3:     **return** *null*

4: **end if**

5: **for** *wi* = 0, . . . ,*w.length* − 1 **do**

6:     $data[wi] = \gamma_s^S (word[wi])$

7: **end for**

8: $S = \{m \mid m[ri : ri + oRow][ci : ci + oCol] \in data[ri \times cnum + ci]\}$

9: *matrix_set* = $\{in \mid in \in \gamma_v(m)$ and $m \in S\}$

10: **return** *matrix_set*

---

word to be concretized should conform to the size of input (Lines 2–4). One can release this length condition by zero-padding or discarding the superfluous symbols. But this may break the Galois connections.

Theoretically, if $(\alpha_v, \gamma_v)$ and $(\alpha_s^S, \gamma_s^S)$ form Galois connections, then so does $(\alpha_w, \gamma_w)$. While for practicality, we use

$$\gamma_w^{\text{prac}} = \gamma_w[\gamma_v \mapsto \gamma_v^{\text{prac}}, \gamma_s^S \mapsto \gamma_s^{Sprac}]$$

instead, and the number of data in $\gamma_w^{\text{prac}}(w)$ depends on the word $w$ as well as $k_v$ and $k_s$. In particular, in our implementation we collect sets of inputs (including values and blocks) that are mapped into an identity word from existing data and then select inputs from the corresponding set.

Finally, considering safety, if $(\alpha_v, \gamma_v)$ and $(\alpha_s^S, \gamma_s^S)$ does not cause the flapping, then neither does $(\alpha_w, \gamma_w)$. But the safety of a network focus on the vibrations of local parts of

inputs (*Huang et al., 2017*). To evaluate the whole inputs, we use another notation *conflict*, that is, inputs of different classifications should not abstracted into an identity word. Formally, we say a network $\mathcal{N}$ is *non-conflict* with respect to $\alpha_w$ and $\gamma_w$ if for every input array *in* and for every array $in' \in \gamma_w(\alpha_w(in))$

$$\mathcal{N}(in') = \mathcal{N}(in)$$

In other words, the abstraction itself should not be over-approximated. We say a word $w$ is *conflict*, if there exist two inputs of different classifications that are abstracted into it. So to avoid over-approximation, the number of the conflict words caused by the abstraction should be as few as possible. Similar to the safety, the non-conflict cannot easily be preserved in practice. For that, we evaluate the conflicts words under a given dataset. In detail, we say a network $\mathcal{N}$ is σ-*conflict* with respect to $\alpha_w$ under a given set $D$ if

$$\frac{|\{in \mid \exists in'.\mathcal{N}(in) \neq \mathcal{N}(in') \wedge \alpha_w(in) = \alpha_w(in')\}|}{|D|} \leq \sigma$$

where $|D| \geq 1$ and $in, in' \in D$.

To sum up, to obtain a suitable abstraction (e.g., scalable, safe and non-conflict), one needs to take into account the number $n$ of intervals, the block size *oRow* and *oCol*, and the other factors.

## Active learning

In this section, we present how to instantiate the active learning framework on neural networks, in particular the membership and equivalence queries.

### *Membership query*

Membership queries can be answered by the neural networks via the word concretization function. In our abstraction, we map a word into a set of data. As mentioned above, the abstraction may flap the results or yield some conflict words, that is, the classifications of different data in the set of an identity word may not be the same. To address this, we count the numbers of different classifications of the data in the set and take the classification which gets the most votes as the result for the word.

Given a network $\mathcal{N}$ and a word concretization function $\alpha_w$, we say a word $w$ is *positive* if

$$|\{in \in \gamma_w(w)|\mathcal{N}(in) = \text{true}\}| \geq |\{in \in \gamma_w(w)|\mathcal{N}(in) = \text{false}\}|$$

and otherwise *negative*. Intuitively, a word is *positive* (*negative* resp.) if there are more positive (negative resp.) input arrays that are abstracted into it than the negative (positive resp.) ones.

Algorithm 3 gives the procedure of membership query checking, where $\mathcal{N}$ denote the neural network under learning. Firstly, the algorithm concretises the word $w$ that is being queried into a set *matrix_set* of possible data, using the word concretization function $\gamma_w$ (Line 1). If the set *matrix_set* is *null*, that is, the length of word $w$ does not conform to the size of input data, then the algorithm returns *false* immediately. Otherwise, the

---

**Algorithm 3** Membership query *M Q(w)*.

---

**Input:** a word *w*

**Output:** true if *w* is accepted, otherwise false

1: *matrix set* = $\gamma_w(w)$

2: **if** *matrix_set == null* **then**

3:     **return** false

4: **end if**

5: *yes* = 0, *no* = 0

6: **for** *matrix in matrix_set* do

7:     **if** $\mathcal{N}$ *(matrix)* **then**

8:         *yes++*

9:     **else**

10:        *no++*

11:     **end if**

12: **end for**

13: **return** *yes >= no*

---

algorithm feeds each data into the neural network $\mathcal{N}$ under learning and counts the numbers of different classifications (Lines 5–12). Finally, it returns the classification that gets the most votes (Line 13).

## Equivalence query

As there is no finite interpretation for neural networks (*Weiss, Goldberg & Yahav, 2018*), equivalence queries are more challenging than membership queries. To address this, similar to *Weiss, Goldberg & Yahav (2018)*'s work, we use an *abstract representation* of the neural network under learning. But different from *Weiss, Goldberg & Yahav (2018)*'s work, we start with the automaton that is learned passively via the RPNI algorithm (*Oncina & Garca, 1992*) from some test queries, which are selected from the training dataset. Then we perform the equivalence query against this abstract model. As discussed in *Weiss, Goldberg & Yahav (2018)*, when a counterexample is found, it may be not that the hypothesis is incorrect, but rather that the abstract model is not precise enough (i.e., different behaviors from the neural network under learning) and needs to be refined.

The procedure[3] of equivalence query checking is given in Algorithm 4. Firstly, the algorithm tries to find a word that can separate the hypothesis $\mathcal{H}$ and the abstract model $\mathcal{M}$ (Line 3). If such a word does not exist, then it returns *null* (Lines 4–6), which means the equivalence query is *yes*. Assume a word *w* is found. Then it checks whether this word is a true counterexample, that is, the classifications of the abstract model and the neural network under learning are the same (Line 7). If it is in that case, then it returns this word as a counterexample to the learner (Line 8). Otherwise, it refines the abstract model with this word (Line 10): it adds the counterexample into the positive set or the

[3] In our implementation, we set a bound for the refining time for efficiency, which may yield an incompatible acceptance exception.

---

**Algorithm 4** Equivalence query $EQ(\mathscr{H}, \mathscr{M})$.

**Input:** a hypothesis $\mathscr{H}$ and an abstract model $\mathscr{M}$

**Output:** a counterexample if $\mathscr{H} \neq \mathscr{M}$, otherwise *null*

1: **while** true **do**
2:     find a word w that separates $\mathscr{H}$ and $\mathscr{M}$
3:     **if** w does not exist **then**
4:         **return** *null*
5:     **end if**
6:     **if** $\mathscr{M}.isAccepted(w) = MQ(w)$ **then**
7:         **return** w
8:     **end if**
9:     refine $\mathscr{M}$ with $(w, MQ(w))$
10: **end while**

---

negative set dependent on its true classification, and relearns a new automata via RPNI. After that, the algorithm continues on the equivalence query against this refined model.

## EXPERIMENTS

We have implemented our approach in a prototype in Java, wherein we use the library *LearnLib* (*Howar et al., 2012*) to implement the MAT learning framework and the RPNI algorithm. Moreover, to find the true counterexamples faster, we use the Wp-method test (*Fujiwara et al., 1991*)[4] in the equivalence query between the hypothesis and the abstract models. To evaluate our approach, we conduct a series of experiments on a classifier for the MNIST dataset, a large database of handwritten digits that is commonly used for training various image processing systems. Firstly, we conduct experiments to see the measures of the MNIST classifier, namely, σ-safety, σ-conflict, the size of alphabet and the length of words, under the abstractions with different interval numbers and block sizes. Secondly, we present the experiments to learn DFAs from the MNIST classifier under different selected abstractions. Thirdly, we also conduct experiments to compare the resulted DFAs against the DFAs learned via the passive learning algorithms provided in LearnLib and the MNIST classifier itself.

The experiments were conducted on a workstation with Intel Processor i7-7820HQ (2.90GHz) and 32GB memory.

### MNIST classifier

The MNIST classifier under learning is a binary classification version of *MnistClassifier* from the tutorial examples of DeepLearning4J (https://github.com/deeplearning4j/dl4j-examples), which recognises the number 1. It is built on a convolution neural network, which consists of six layers, namely, a convolution layer, a pooling layer, another convolution layer, another pooling layer, a dense layer and an output layer. The training

[4] The other equivalence approximation strategies provided in LearnLib can be used as well.

**Table 1 Flapped results on different $k$-value manipulations.**

| $n$ | $k$ | Flaps | Ratio (%) | $n$ | $k$ | Flaps | Ratio (%) |
|---|---|---|---|---|---|---|---|
| 2 | 1 | 3 | 0.005 | 2 | 10 | 6 | 0.010 |
| 2 | 100 | 32 | 0.053 | 2 | 500 | 276 | 0.461 |
| 3 | 1 | 0 | 0.000 | 3 | 10 | 5 | 0.008 |
| 3 | 100 | 16 | 0.027 | 3 | 500 | 43 | 0.072 |
| 5 | 1 | 0 | 0.000 | 5 | 10 | 0 | 0.000 |
| 5 | 100 | 10 | 0.017 | 5 | 500 | 16 | 0.027 |
| 10 | 1 | 1 | 0.002 | 10 | 10 | 0 | 0.000 |
| 10 | 100 | 5 | 0.008 | 10 | 500 | 6 | 0.015 |

dataset and the testing dataset are from the official site (http://yann.lecun.com/exdb/mnist/), wherein each input is 2-dimensional integer matrix with size $28 \times 28$.

## Abstraction experiments

As discussed in "Abstraction", the interval number $n$ and the block size affect the definition of the abstraction, especially the safety and the conflict of the neural network under learning, the size of alphabet and the length of words. For that, we present in this section some experiments to see these measures of the abstractions with different interval numbers and block sizes.

### Safety

The first measure to test is the safety. For that, we present some experiments to test the flapping of the MNIST classifier on some selected inputs from the training set (i) via performing $k$-value manipulations $vm^k$ with different interval numbers and (ii) via performing $k$-shift manipulations $sm^k$ with different block sizes.

First, in the experiments about $k$-value manipulation $vm^k$, for a given interval number $n$, we randomly select $k$ values from a selected input, and replace each selected value by a random value which shares the same interval with the corresponding selected value. Then we fed the resulted data into the MNIST classifier and see whether the classifications are flapped. We select 59,838 inputs in total from the training set, which are classified correctly by the MNIST classifier. Table 1 shows the results, where **Flaps** denotes the number of inputs whose results are flapped by the manipulation, and **Ratio** denotes the percentage of the number of flapped input to the total number of selected inputs.

From the results we can see that, the number of flapped inputs increases as the number $k$ of selected values increases, since the larger the number $k$, the larger the vibration for the inputs. In contrast, as the number of intervals increase, the number of flapped inputs decreases, which indicates that the larger the interval number, the better. This conforms to the discussion in "Abstraction". Moreover, the results also show that the MNIST classifier is about 0.053%-safety, with respect to the 100-value manipulation $vm^{100}$ with the interval number 2. And the 100-value manipulation means 12.76% (100/784) of an input has been modified, such that we believe 100 is enough for local vibration. Therefore, we suggest to set the interval number $n$ as 2.

**Table 2 Flapped results on _k_-shift manipulations.**

| oRow | oCol | k | Flaps | Ratio (%) | oRow | oCol | k | Flaps | Ratio (%) |
|------|------|---|-------|-----------|------|------|---|-------|-----------|
| 1 | 28 | 1 | 0 | 0.000 | 28 | 1 | 1 | 4 | 0.007 |
| 1 | 28 | 4 | 33 | 0.055 | 28 | 1 | 4 | 376 | 0.628 |
| 1 | 28 | 28 | 5136 | 8.583 | 28 | 1 | 28 | 6608 | 11.043 |
| 2 | 28 | 1 | 10 | 0.017 | 28 | 2 | 1 | 105 | 0.175 |
| 2 | 28 | 2 | 109 | 0.182 | 28 | 2 | 2 | 879 | 1.469 |
| 2 | 28 | 14 | 5768 | 9.639 | 28 | 2 | 14 | 6615 | 11.055 |
| 4 | 28 | 1 | 382 | 0.638 | 28 | 4 | 1 | 1018 | 1.701 |
| 4 | 28 | 7 | 6284 | 10.502 | 28 | 4 | 7 | 6617 | 11.058 |
| 7 | 28 | 1 | 1273 | 2.127 | 28 | 7 | 1 | 2077 | 3.471 |
| 7 | 28 | 4 | 6684 | 11.170 | 28 | 7 | 4 | 6620 | 11.063 |
| 14 | 28 | 1 | 3050 | 5.097 | 28 | 14 | 1 | 3766 | 6.294 |
| 14 | 28 | 2 | 6436 | 10.756 | 28 | 14 | 2 | 6431 | 10.747 |
| 28 | 28 | 1 | 6425 | 10.737 | – | – | – | – | – |

Next, in the experiments about $k$-shift manipulation $sm^k$, for a given block size $oRow \times oCol$, we randomly select $k$ blocks from a selected input, and rearrange the values in each selected block. Then we feed the resulted data into the MNIST classifier and see whether the classifications are flapped. Similarly, we select the 59,838 inputs that are classified correctly by the MNIST classifier from the training set. For simplicity and scalability, we consider the block sizes whose row sizes or column sizes are 28. The results are given in Table 2, where the notations are the same as the ones in Table 1.

First, the results show that, as the size of block increases, the number of flapped inputs increases, which conforms to the discussion in "Abstraction". The results also show that the number of flapped inputs increases as the number $k$ of selected blocks increases. This is because that, the larger the number $k$, the larger the vibration for the inputs. Moreover, we found that the MNIST classifier is more safe under $k$-shift manipulation built on rows than the one on columns. The reason may be that the digit number of 1 is more regular in row order than in column order. Finally, assume the size allowed for local vibration is about 100. All the σ-safeties of the MNIST classifier with respect to the $k$-shift manipulation with the block size $1 \times 28$, $2 \times 28$, $4 \times 28$, $28 \times 1$, $28 \times 2$ or $28 \times 4$ are smaller than 1.8%. In particular, the MNIST classifier is about 0.055%-safety, with respect to the 4-shift manipulation $vm^4$ with the block size $1 \times 28$.

## Conflict

The second measure to test is the non-conflict, which indicates whether the abstraction with the given block size is over-approximated. In other words, we would like to conduct experiments to test how many conflict words that are generated by the abstractions with different block sizes under the training set. For that, we perform the abstractions with different block sizes on some selected inputs from the training set, and do a statistic analysis on the abstracted words with respect to their classifications, where we take the

**Table 3 Conflict results on different abstractions.**

| oRow | oCol | TW | PW | NW | CPW | CNW | CPD | CND |
|------|------|-------|------|-------|-----|-----|------|------|
| 1 | 28 | 59745 | 6605 | 53140 | 0 | 0 | 0 | 0 |
| 2 | 28 | 59640 | 6500 | 53140 | 0 | 0 | 0 | 0 |
| 4 | 28 | 58817 | 5755 | 53062 | 15 | 0 | 0 | 15 |
| 7 | 28 | 53912 | 4501 | 49411 | 246 | 11 | 11 | 248 |
| 14 | 28 | 5444 | 248 | 5196 | 163 | 544 | 2577 | 1367 |
| 28 | 1 | 59708 | 6569 | 53139 | 1 | 0 | 0 | 1 |
| 28 | 2 | 59321 | 6195 | 53126 | 8 | 0 | 0 | 8 |
| 28 | 4 | 55944 | 4512 | 51432 | 421 | 80 | 81 | 431 |
| 28 | 7 | 36535 | 1414 | 35121 | 490 | 744 | 2276 | 1333 |
| 28 | 14 | 5229 | 734 | 4495 | 327 | 871 | 3196 | 1195 |
| 28 | 28 | 239 | 37 | 202 | 24 | 79 | 3765 | 1204 |

interval number $n$ as the suggested one 2. The test inputs that are selected from the training set is 59,840 in total, with 6,700 positive inputs and 53,140 negative ones.

The statistic results are given in Table 3, where **TW** denotes the total number of words, **PW** (**NW** resp.) denotes the number of positive (negative resp.) words, **CPW** (**CNW** resp.) denotes the number of positive (negative resp.) words that have both positive and negative inputs and **CPD** (**CND** resp.) denotes the number of positive (negative resp.) inputs that are abstracted into a negative (positive resp.) word.

The results show that as the block size increases, the number of abstracted words decreases, which conforms to the discussion in "Abstraction". Thus it could be easier to extract the automaton for a larger block size. For example, when taking the whole input as a symbol, there are 239 words in total. But both the number of conflict words and the number of conflict data increase as the block size increases, which indicates that an abstraction with a larger block size is prone to be an over-approximation. In particular, when taking the whole input as a symbol, 56.194% of the positive inputs are abstracted into negative words and 64.865% of the positive words are conflict. Moreover, from the results we can also see that all the σ-conflicts for the MNIST classifier with respect to the abstractions with the block size 1 × 28, 2 × 28, 28 × 1 or 28 × 2 are smaller than 0.015% (8/59,840). And the abstractions with the block size 2 × 28 and 1 × 28 perform best, yielding none conflict data nor words.

### Word complexities

Finally, we also conduct experiments to see the size of alphabets and the length of words, which are dubbed as word complexities. Table 4 shows the word complexities under different abstractions with different block sizes, where **Size** denotes the size of alphabet, **dSize** denotes the number of symbols occurring in the selected inputs and **Length** denotes the length of words.

The results show that the larger the block size, the larger the alphabet size and the shorter the word length, which conforms to the discussion in "Abstraction". Moreover, we found that the products of the alphabet size and the word length are almost the same.

**Table 4 Word complexities on different abstractions.**

| n | oRow | oCol | dSize | Size | Length | n | oRow | oCol | dSize | Size | Length |
|---|------|------|-------|------|--------|---|------|------|-------|------|--------|
| 2 | 1 | 28 | 21 | 29 | 28 | 2 | 28 | 1 | 21 | 29 | 28 |
| 2 | 2 | 28 | 41 | 57 | 14 | 2 | 28 | 2 | 41 | 57 | 14 |
| 2 | 4 | 28 | 48 | 113 | 7 | 2 | 28 | 4 | 81 | 113 | 7 |
| 2 | 7 | 28 | 126 | 197 | 4 | 2 | 28 | 7 | 124 | 197 | 4 |
| 2 | 14 | 28 | 143 | 393 | 2 | 2 | 28 | 14 | 146 | 393 | 2 |
| 2 | 28 | 28 | 244 | 785 | 1 | – | – | – | – | – | – |

**Table 5 Automata performance under different abstractions.**

| oRow | oCol | wPre (%) | wRec (%) | wAcc (%) | wF1 (%) | dPre (%) | dRec (%) | dAcc (%) | dF1 (%) |
|------|------|----------|----------|----------|---------|----------|----------|----------|---------|
| 1 | 28 | 55.041 | 70.782 | 90.176 | 61.927 | 55.288 | 70.988 | 90.184 | 62.162 |
| 2 | 28 | 54.830 | 70.321 | 90.142 | 61.617 | 55.203 | 70.635 | 90.154 | 61.973 |
| 28 | 1 | 44.382 | 70.035 | 86.691 | 54.333 | 44.569 | 70.194 | 86.699 | 54.521 |
| 28 | 2 | 42.904 | 69.686 | 86.232 | 53.110 | 43.466 | 70.106 | 86.248 | 53.662 |
| 28 | 28 | 100.000 | 66.667 | 94.064 | 80.000 | 74.472 | 40.388 | 91.657 | 52.373 |

So for scalability, any block size seems fine. But if considering the practical alphabet (i.e., symbols occurring in the inputs), the larger block size could be better.

To sum up, based on the experiments above, we suggest to use for the MNIST classifier the abstractions with the interval number $n = 2$ and the block size $1 \times 28$, $2 \times 28$, $28 \times 1$ or $28 \times 2$.

## Automata learning

In this section, we present the experiments to learn DFAs from the MNIST classifier under the suggested abstractions.

To quantitatively validate the models, we use the following performance measures. *Accuracy* is the most intuitive performance measure and it is simply a ratio of correctly predicted observation to the total observations. *Precision* is the ratio of correctly predicted positive observations to the total predicted positive observations, and *Recall* is the ratio of correctly predicted positive observations to all observations in actual class. *F1 score* is the weighted average of *Precision* and *Recall*, that is, $(2 \cdot Precision \cdot Recall)/(Precision + Recall)$. Moreover, there are two kinds of observations in our experiments, namely, the input arrays and the abstracted words (i.e., the abstractions of the input arrays). So we compute the measures above with respect to both kinds of observations. Intuitively, the higher the measures above, the better the model.

### Automata performance

We first conduct experiments to learn DFAs from the MNIST classifier under different suggested abstractions. Then we perform experiments to evaluate the learned DFAs on the testing dataset. The performance results of the learned DFAs are given in Table 5, where the interval number $n$ is 2, the columns **wAcc**, **wPre**, **wRec** and **wF1** respectively

 

**Table 6 Learning complexities under different abstractions.**

| oRow | oCol | aTime | xTime | iTime | aState | xState | iState |
|---|---|---|---|---|---|---|---|
| 1 | 28 | 29150.5 | 47761.8 | 16393.2 | 579 | 949 | 251 |
| 2 | 28 | 8145.3 | 15833.9 | 2838.4 | 299.3 | 495 | 139 |
| 28 | 1 | 16164.0 | 65525.8 | 75.7 | 801.1 | 2213 | 239 |
| 28 | 2 | 6492.4 | 12650.5 | 214.6 | 309.3 | 499 | 97 |
| 28 | 28 | 5.9 | 11.8 | 0.1 | 2 | 2 | 2 |

denote the *Accuracy*, *Precision*, *Recall* and *F1 score* that are computed with respect to words, and the columns **dAcc**, **dPre**, **dRec** and **dF1** respectively denote the *Accuracy*, *Precision*, *Recall* and *F1 score* that are computed with respect to input data.

The results show that all the learned DFAs perform well on the testing dataset, with the *F1 score* more than 50%[5]. In particular, the DFAs learned under the abstraction with block size 1 × 28 performs best. The results also show that DFA learned via the abstraction with a smaller block size can obtain a higher *F1 score*. In detail, the *F1 score* of DFA learned via the abstraction with block size 1 × 28 (28 × 1 resp.) is higher than the one with block size 2 × 28 or 28 × 2 (28 × 2 resp.). This is because that a smaller block size can generate a more preciser abstraction, which conforms to the discussion in "Abstraction". Moreover, from the results, we can see that the DFAs learned under the abstractions in rows perform better than the ones under the abstractions in columns in terms of all performance measures with respect to both words and data. For example, the *F1 score* of DFA learned via the abstraction with block size 1 × 28 is higher than the one with block size 28 × 1. The reason may be that the digit number of 1 is more *regular* in row order than in column order. In addition, we also perform the abstraction mapping a whole input as a symbol as does in *Weiss, Goldberg & Yahav (2018)*'s work. Due to this abstraction is over-approximated, the extracted DFA gets the worst performance in the input data layer, although it has a better performance than the other models on the word layer, especially the *Precision*.

### Learning complexities

During the experiments, we also count the learning times in seconds needed by the resulted DFAs and the sizes of the resulted DFAs. The results are given in Table 6, where **aTime**, **xTime** and **iTime** respectively denote the average time, the maximum time and the minimum time needs by the resulted DFAs, and **aState**, **xState** and **iState** respectively denote the average number, the maximum number and the minimum number of states of the resulted DFAs.

From the results, we can see that learning the DFA via a smaller block size needs more time. As discussed before, an input array can be abstracted into a longer word under the abstraction with a smaller block size, which thus requires more time to proceed. Concerning the size of learned DFA, learning via a smaller block size can yield a larger DFA. For example, the number of the states of the DFA learned under the abstraction with block size 28 × 1 is the largest one among the results. Similar to the learning time, the reason is that an abstraction with a smaller block size yields longer words, which could

[5] Assume a random binary classifier predicts half of the digits as 1 with the accuracy 50%. As the ratio of digit 1 in the dataset is about 1/10, the F1 score is about 16.7%.

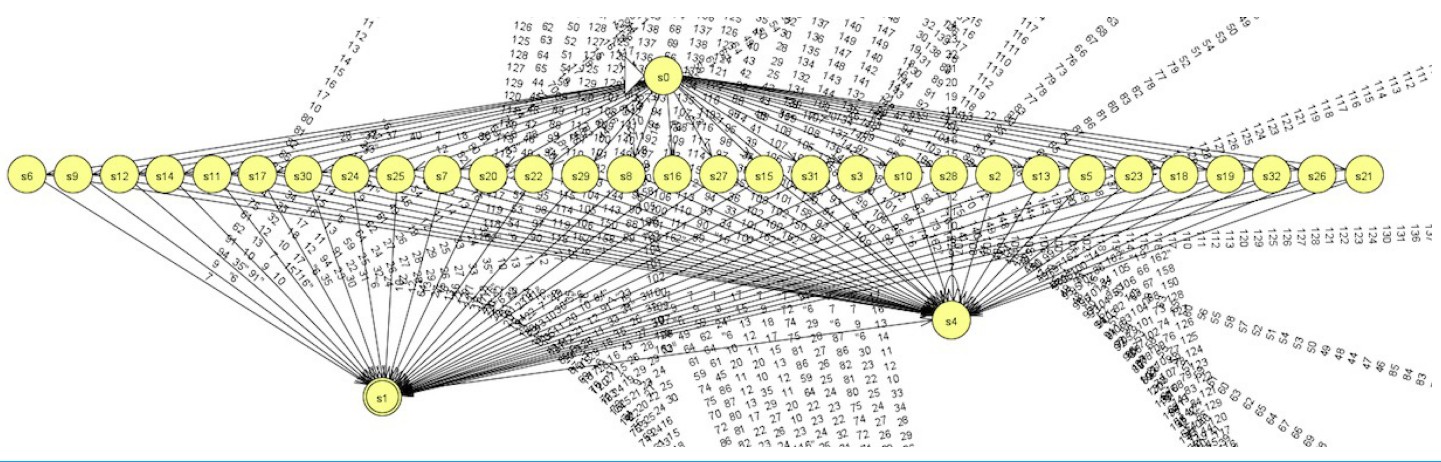

**Figure 10 DFA learned via abstraction with 14 × 28.**

enlarge the learned DFA. In addition, both the learning times and the sizes of the learned DFAs under the abstractions in rows are larger than the ones under the abstractions in columns. The reason may be that more blocks are abstracted into 0 under the abstractions in columns than the abstractions in rows.

### Learned automata

Finally, we convert the learned DFAs into the format used in *JFLAP* (*Rodger & Finley, 2006*), which enables us to view the DFAs and convert DFA into regular expressions step by step. For simplicity, we consider a DFA learned via the abstraction with block size 14 × 28, which is given in Fig. 10. From this figure, we can see that (i) the learned DFA has a very clear structure: a starting node, an intermediate layer with several nodes, an accepting node and a trap node; and (ii) the learned DFA accepts words with length 2. To understand it further, we convert this DFA into a regular expression, which is a union of several alternatives. Among these alternatives, some of them are easy to understand and explain. For example, the expressions $(31 + 29)(31 + 32 + 30 + 29 + 33)$, $25(23 + 31 + 26 + 27 + 28 + 29 + 24 + 25 + 30)$ and $26(30 + 31 + 32 + 26 + 27 + 28 + 29 + 25 + 33 + 24 + 34)$ state that the upper part and the low part share similar sums.

Let us see a DFA learned via the abstraction with block size 7 × 28, which is shown in Fig. 11. Compared with the one in Fig. 10, this DFA has a more complex structure. But we still can identity some hierarchical structures in it. It is pity that we are not able to convert this DFA into a regular expression via JFLAP due to a runtime error.

In addition, we also present a DFA learned via the abstraction with block size 1 × 28, which has 949 states and is given in Fig. 12. It is a little complex to understand, so we can only identity a rough hierarchical structure. We believe that one can understand this DFA more if he gets a closer look on it. In addition, the learned DFAs can help to generate test cases to test the networks, which are left as a future work.

### Comparison

To further evaluate the resulted DFA, we compare it against the DFAs learned via the passive learning algorithms provided in LearnLib, namely, the RPNI algorithm, the RPNI-

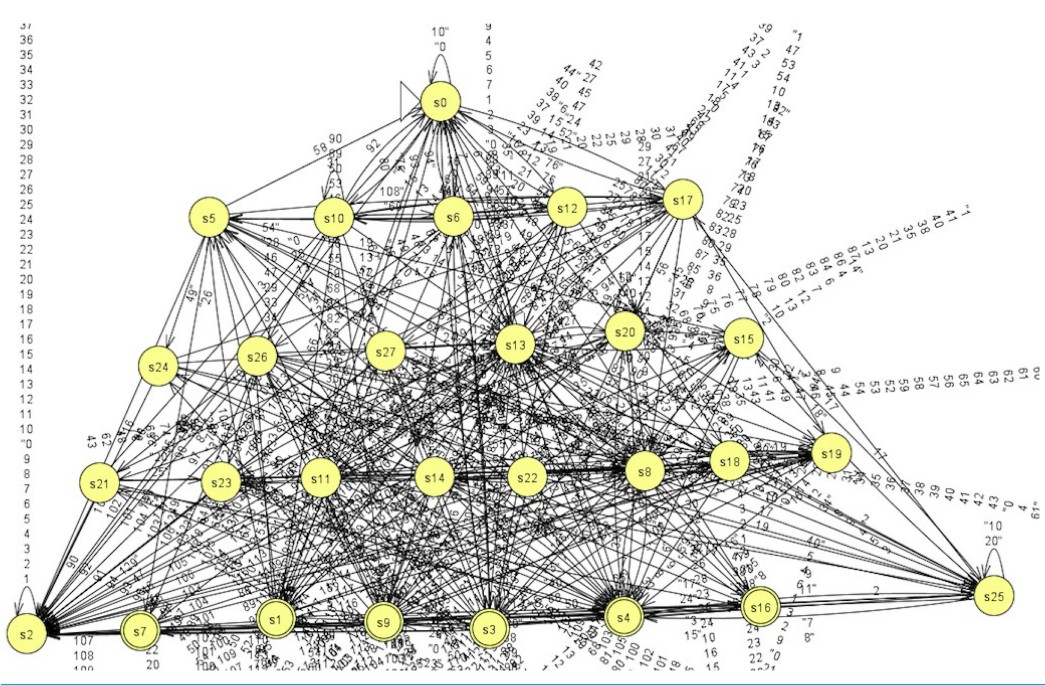

**Figure 11 DFA learned via abstraction with 7×28.**

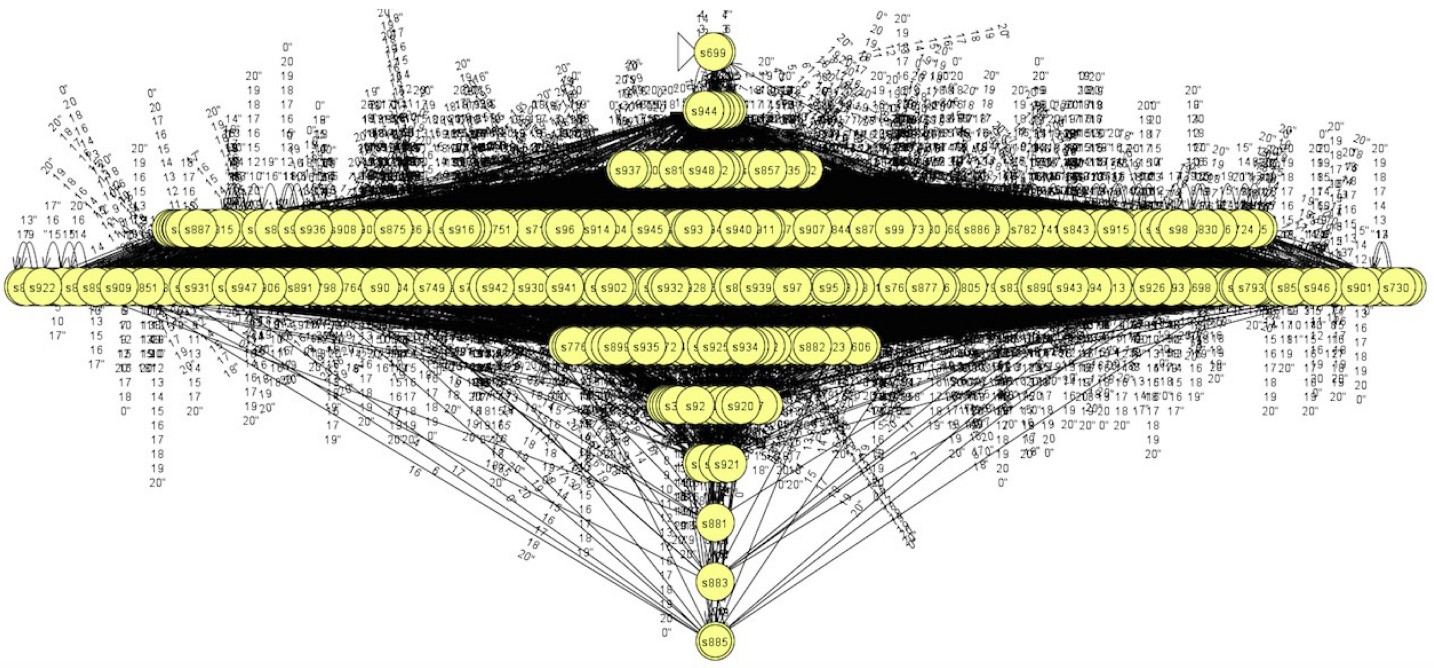

**Figure 12 DFA learned via abstraction with 1 × 28.**

EDSM algorithm and the RPNI-MDL algorithm, and the MNIST classifier itself. In these experiments, we perform our abstraction on the training data and then learn a DFA via each passive learning algorithms provided in LearnLib, wherein the abstraction we used is the one with interval number 2 and block size 1 × 28, all the arrays are selected for

**Table 7 Comparison against the passive DFAs and the MNIST classifier.**

| Model | wPre (%) | wRec (%) | wAcc (%) | wF1 (%) | dPre (%) | dRec(%) | dAcc (%) | dF1 (%) |
|---|---|---|---|---|---|---|---|---|
| Ours | 55.041 | 70.782 | 90.176 | 61.927 | 55.288 | 70.988 | 90.184 | 62.162 |
| RPNI | 46.170 | 59.947 | 87.590 | 52.164 | 46.463 | 60.229 | 87.600 | 52.458 |
| RPNI-MDL | 15.599 | 99.911 | 38.974 | 26.985 | 15.693 | 99.912 | 39.022 | 27.125 |
| RPNI-EDSM | 19.407 | 98.313 | 53.729 | 32.416 | 19.520 | 98.325 | 53.766 | 32.574 |
| CNN | – | – | – | – | 99.69 | 99.40 | 99.06 | 98.63 |

the RPNI algorithm, only the positive arrays are selected for the RPNI–MDL algorithm (since it does not support the negative examples), and all the positive arrays and only one ninth of the negative ones are selected for the RPNI–EDSM algorithm (to avoid memory overflow and time-consuming). Next, we evaluate all the models with the testing dataset. The results are given in Table 7, where the notations are the same as the ones of Table 5.

Compared to the RPNI one, our DFA performs better in all the performance measures, since the abstract model we use is the DFA learned via the RPNI algorithm from some inputs in the training dataset and is refined with respect to the classifier continually during learning. While compared to the RPNI–MDL and RPNI–EDSM ones, our DFA has a better *Accuracy*, *Precision* and *F1 score*, but a worse *Recall*. This is because that these two DFAs take all the positive inputs in the training set into account such that it can recognise more positive inputs in the testing dataset, while only part of positive inputs are selected for our abstracted model. The results also show that our DFA is still worse than the classifier. There are several reasons for this. The first one is that we have set some bounds (e.g., the refining time for the abstract representation) in our implementation for the learning procedure for efficiency and to avoid memory overflow. The second one is that the Wp-method test used in our experiments may miss some true counterexamples. The third one is that the abstraction may be over-approximated to yield too many conflict words. Nevertheless, our approach still needs to be improved.

## LIMITATIONS

Although our approach works for the MNIST classifier, there are still some limitations. Firstly, to figure out a suitable abstraction for the neural network under learning is not an easy task. As shown in *Biggio et al. (2013)*, *Szegedy et al. (2013)* and *Huang et al. (2017)*, several DNN, including highly trained and smooth networks optimised for vision tasks, are unstable with respect to so called *adversarial perturbations*. Hence, some neural networks may be too sensitive to the abstraction manipulation to find a reasonable interval number. Even if a reasonable interval number were found, one need to make a compromise between the abstraction and the scalability to find a block size. Moreover, whether a turing machine can simulate a natural neural network is an open question (*Zenil & Quiroz, 2006*). So in some sense, we cannot define an abstraction without the conflict or the flapping.

Secondly, the scalability is another problem. Generally, the size of inputs of neural networks is in thousands. For such a neural network, either the alphabet may be too large

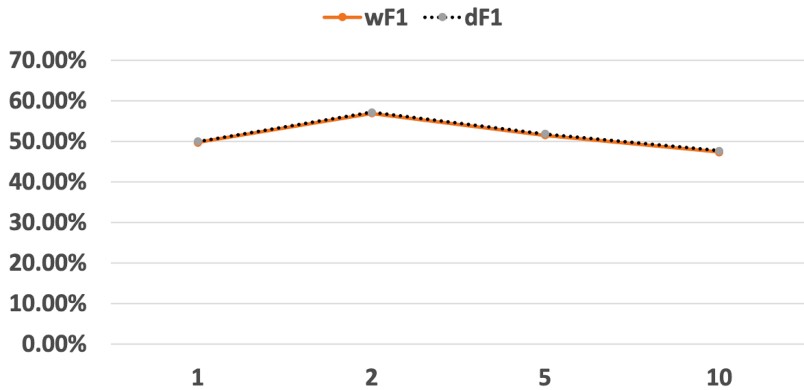

**Figure 13** The *F1* scores with respect to words and input data via the abstraction 2 × 28.

(if a large block size is taken) or the word may be too long (if a small block size is taken) for us to extract the automaton. Taking the MNIST classifier for example, it may last several hours for some abstractions (see the abstraction with block size 1 × 28 in Table 6) to extract the automaton, even several days. There are two possible reasons for this issue in our implementation: (i) we use the implementation of RPNI algorithm from LearnLib, which does not support incremental learning proposed in *Dupont (1996)* and can be improved with it; and (ii) there are too many queries for the non-accepting words with invalid lengths.

Thirdly, our approach is dependent on the dataset. In "Experiments", we selected the interval number and the block size via an analysis on the training dataset. Different datasets may derive different abstractions. To make things worse, it may be the case that an abstraction is suitable for the training dataset, but unsuited for some other testing dataset. Moreover, our abstract model is built from some existing testing data. Different data yields different abstract models, which could affect the results, such as the learning time and the learned DFA.

Fourthly, the implementation of equivalence query is a practical problem requiring attention. One may think that a precise equivalence check can be performed in polynomial time on the hypothesis automaton and the abstract automaton. However, the precise equivalence check could return too many false counterexamples such that it takes too much time for the learning. This is because the precise equivalence check is prone to generate a short and false counterexample that is invalid with respect to the abstraction. Indeed, we have tried this precise equivalence check, but we only succeed on the abstraction of 28 × 28 in 1 h. So considering the efficiency, we use the Wp-method test in the equivalence check, which enables us to find the counterexamples whose lengths are in a given range.

Fifthly, although our approach is black-box, the structures of neural networks may affect the performances of the networks themselves, so as the performances of the learned DFAs. We have performed our approach with the abstraction 2 × 28 on the MNIST classifiers whose hidden layer numbers range in {1, 2, 5, 10}, which are binary classification versions from the tutorial examples of DeepLearning4J. The accuracies of all the classifiers are above 99%. Figure 13 shows the *F1* scores with respect to words and input data of

the learned DFA. We can see that the *F*1 scores are quite close to each other. And the DFA learned from the larger network is not necessary to get the best performance in terms of *F*1 score. Therefore, we present the experimental results on only one MNIST classifier in "Experiments".

## RELATED WORK

In this section, we review some related work. Existing work on DFA extraction from neural networks targets RNNs, which was extensively explored in *Jacobsson (2005)* and *Wang et al. (2017)*.

*Omlin & Giles (1996)* proposed a global partitioning of the network state space according to *q* equal intervals along every dimension, and then exploring the network transitions in the partitioned space. Our value abstraction adopts this partitioning, but we work on the input space, instead of the state space.

*Cechin, Simon & Stertz (2003)* presented a approach to extract DFA using *k*-means and fuzzy clustering. The key idea is to classify a large sample set of reachable network state using *k*-means. *Hou & Zhou (2018)* proposed another approach to extract DFA from RNN using two clustering algorithms, namely LISOR-k and LISOR-x, on hidden states. There are several other work that adopted cluster analysis on state space, including *k*-means clustering (*Zeng, Goodman & Smyth, 1993*; *Frasconi et al., 1996*; *Gori et al., 1998*; *Cohen et al., 2017*), hierarchical clustering (*Sanfeliu & Alquezar, 1994*) and self-organizing maps (*Tiňo & Šajda, 1995*). These approaches have to access the state-vectors, while our approach is a black-box one.

Recently, *Weiss, Goldberg & Yahav (2018)* adopted active learning to extract automata from RNN. Our work is inspired by and similar to this, but different in the follows: (1) we target general neural network, not only RNN; (2) we consider an input is a word, rather than a symbol; (3) we use a DFA that is inferred from some training data as an abstract model for equivalent queries.

## CONCLUSION

In this work, we have proposed a MAT framework to extract automata from neural networks, employing abstraction interpretation of the neural networks for answering membership and equivalence queries. We have implemented our approach in a prototype and have carried out some interesting experiments on a MNIST classifier. Through experiments, we have found that the DFA extracted from the MNIST classifier under the abstraction with the interval number 2 and the block size $1 \times 28$ performs the best. In the experiments, that our resulted DFA has a better performance than the DFAs learned via the passive algorithms provided in LearnLib on the MNIST dataset.

As for future work, we may consider a better encoding such as RLE to improve the approach. We can improve the RPNI algorithm with incremental learning to reduce the learning time. We can also perform experiments on other neural network classifiers. Other models to be extracted from neural network are under consideration.

### Funding

This work was supported by the National Natural Science Foundation of China (Nos. 61972260, 61772347, 61836005) and the Guangdong Basic and Applied Basic Research Foundation (No. 2019A1515011577). The funders had no role in study design, data collection and analysis, decision to publish, or preparation of the manuscript.

### Grant Disclosures

The following grant information was disclosed by the authors:
National Natural Science Foundation of China: 61972260, 61772347 and 61836005.
Guangdong Basic and Applied Basic Research Foundation: 2019A1515011577.

### Competing Interests

Shengchao Qin is an Academic Editor for PeerJ Computer Science.

### Author Contributions

- Zhiwu Xu conceived and designed the experiments, performed the experiments, analyzed the data, performed the computation work, prepared figures and/or tables, authored or reviewed drafts of the paper, and approved the final draft.
- Cheng Wen conceived and designed the experiments, performed the experiments, analyzed the data, performed the computation work, prepared figures and/or tables, and approved the final draft.
- Shengchao Qin conceived and designed the experiments, authored or reviewed drafts of the paper, and approved the final draft.
- Mengda He analyzed the data, prepared figures and/or tables, and approved the final draft.

### Data Availability

The data, code and jar files are available at Figshare:
xu (2021): jars. figshare. Software. DOI 10.6084/m9.figshare.14132531.v1.
xu (2021): data. figshare. Dataset. DOI 10.6084/m9.figshare.14132507.v1.
xu (2021): code. figshare. Software. DOI 10.6084/m9.figshare.14132393.v1.
Data and code are also available in the Supplemental Files.

### Supplemental Information

Supplemental information for this article can be found online at http://dx.doi.org/10.7717/peerj-cs.436#supplemental-information.

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
