# Peer review of "Extracting automata from neural networks using active learning"

_PeerJ Computer Science, doi:10.7717/peerj-cs.436_

## Round 0.1 · original submission · Major Revisions

The reviewers see clear merits in the paper and think that the general idea of the paper is interesting and is apparently sound.

They point out issues in the presentation and also in the experimental design of the evaluation, listed in the reviewer's comments, which need to be considered in the forthcoming major revision of the paper.

Reviewer 1 ·

Basic reporting

Positive:
+ Embedding into existing research landscape well-described.
+ Relevant literature is covered at appropriate depth.
+ Approach idea clearly described.
+ Raw data shared.
+ Self-contained research contribution, including proposal of a new technique and its evaluation.
+ The writing is generally good.

Needs improvement:
- The text contains a certain number of grammar errors. These typically do not impair understanding, but make the reading of the article more tedious. For examples, see my annotated PDF, which also contains a few additional hints.

Experimental design

Positive:
+ Fits into the scope of PeerJ CS
+ Appropriately points out cap in existing knowledge
+ Investigation sufficiently rigorous
+ Sufficient detail for replication
+ Relevant, well-defined research questions

Needs improvement:
- While generally self-contained, the article does not explain the choice of the comparative approach for evaluation (passive learning). Specifically, it does not explain how this approach works, and why it is a suitable comparator for evaluation.
- The evaluation was performed on only one case. The considered case was chosen because it represents a typical "hello world" program for machine learning. There needs to be a discussion why not a more realistic neutral network was considered, and why only one.

Validity of the findings

Positive:
+ Underlying data have been provided
+ Discussion of limitations very welcome

Needs improvement:
- Since the evaluation is based on a single example, chosen due to its use as a "hello world" case, the conclusions about performance in abstract/introduction/conclusion have to be toned down. It's OK to summarize the results as positive or promising, especially with regard to the considered case. But it's somewhat far-fetched to say "we have confirmed that our DFA gives a better performance" without qualifications.
- I am not sure whether I can agree to the following statement: "The results show that all the learned DFAs perform well on the testing dataset, with the F1 score more than 50%." For a binary classifier, a F1 score close to 50% indicates a close-to-random performance. Please either explain this apparent contradiction, or weaken down the statement (note that negative/inconclusive results are accepted).
- In Section 4.4, the text on the comparison to the comparator tools needs to include more details and qualifications (also hard to see in table). How much better was the new approach? It might be a good idea to perform a statistical analysis (although not strictly requirement if performance gain is obvious).

Additional comments

I have some doubts about the general usefulness of the approach, notably this circumstance: "Note that,
when a counterexample is found, it may be not that the hypothesis is incorrect, but rather that the abstract model is not precise enough and needs to be refined." Without a way to tell both sources of errors apart, it might be hard derive meaningful information about whether the produced models are correct. Maybe the authors can comment on their ideas to address this concern.

However, since this issue is more about the usefulness/importance of the results rather than soundness, it did not inform my recommendation.

Annotated reviews are not available for download in order to protect the identity of reviewers who chose to remain anonymous.

Reviewer 2 ·

Basic reporting

The article discusses the application of active automata learning for
neural network classifiers. A deterministic finite automaton is
extracted from a trained network using the OSS library LearnLib. The
purpose is to better understand what is going on in the black-box of
the ML component. The evaluation is based on a MNIST classifier for
handwritten digits. Different possible abstractions in order to apply
the L*-algorithm are proposed and evaluated. At the end visual
representation of the learned automata are presented and discussed.

Regarding presentation, the paper is well structured and easy to
read. I especially appreciated the examples guiding through the
different abstraction steps. The explanations regarding the
experiments should be improved as it is not always clear what data is
presented (see details below).

Experimental design

As the authors state, the idea of extracting an automata from a
trained neural network is not new. However, the field is only at its
beginning and the authors do not assume access to the internals of the
network and evaluate a few new ideas regarding the abstraction of
images. Hence, the work is novel and interesting. The main
contribution is the three-layered abstraction that enables the
processing of the images. Although, the selected images are rather
small.

For equivalence testing, I find the use of an automata, passively
learned from the training data, interesting. However, a weakness of
the approach is the use of testing for comparing the two automata. The
testing algorithms as offered by Learnlib are meant for implementing
equivalence oracles for black-box systems. However, you have both
automata available and can use a precise equivalence check on the
automata which can be done in polynomial time. As a result of using
testing, you may miss more counterexamples as necessary and your
learned automata may be inaccurate.

Validity of the findings

The techniques and the evaluation are sound. Standard algorithms from
LearnLib are used and the abstractions also make sense. The evaluation
uses a standard benchmark and analyses the data from different
perspectives which gives interesting insights into the application of
L* for classification of images.

Additional comments

Hence, the strength of this work is certainly its novelty and the work on abstraction. A weakness is the
implementation of the equivalence oracle. Since I consider the points in favour as being more essential
I recommend publication.

Details:
L10 and 26: "Deep Learning is a new area of Machine Learning research". You start your article with a false statement. It is not new, but around since a while now.

L35: "and what they eventually learn in actuality" ==> and what they eventually learn actually

L43: There is also a newer survey that should be cited:
Model learning and model-based testing. BK Aichernig, W Mostowski, MR Mousavi ,
M Tappler, M Taromirad, Machine Learning for Dynamic Software Analysis: Potentials and Limits, 74-100

L48: symbol data ==> symbolic data

L195: It is the same sigma-safety formula than above. Replace, vm^k with sm^k.

L302: "And we believe 100 is enough for local vibration, so we suggest to set the interval number n as 2." Is it?
To me it is not clear how you support this believe. It could as well be that choosing a different interval number could result in more accurate automata. Did you try learning, e.g. with n=3?

L279: a output ==> an output

Table 3: Please, also provide a column with the number of total words, as you discuss how the total number decreases with
increasing block size.

L363: Please, provide a more explicit explanation of the difference between wAcc, ... and dAcc. How exactly are these
calculated. I did not immediately understand what "with respect to words" and "with respect to input data" means.
However, after some thought I figured it out, but you should help the reader here. Otherwise, the rest of the
discussion is lost.

L430-432: this explanation is a bit vague. What bounds are you referring here to? Is it just an implementation issue, i.e.
incomplete learning or is it due to the abstraction? The equivalence oracle may be another cause. It is only based on
the training data and not on the trained network. Please, extend this discusion/conclusion, because it is quite
relevant how your approach relates to the true CNN.

L435: a easy task ==> an easy task

L446: for for ==> for

---

## Round 0.2 · Minor Revisions

The only outstanding issues is the availability of the data files for the sake of reproducibility. Once this is made available and is commented upon in the paper and the rebuttal, I am happy to recommend acceptance.

Reviewer 1 ·

Basic reporting

All of my issues pointed out have been addressed satisfactorily.

In addition, beyond my initial check of the provided raw data, I had a closer look at the provided raw data file.

On the positive side, I can verify that the file contains the implementation source code.
Towards a possible improvement, there are two issues:
(1.) Usage instructions for the artifact are missing, I tried compiling the main projects as a maven build, but that lead to an error (see below).
(2.) Evaluation data (measurements) are either missing or not easy to find.
Both issues should be fixed for the final version.


C:\Users\XXX\Downloads\cs-54449-dfa-mnist\dfa-mnist>mvn clean build
[INFO] Scanning for projects...
[ERROR] [ERROR] Some problems were encountered while processing the POMs:
[FATAL] Non-resolvable parent POM for org.deeplearning4j:dfa-mnist:1.0.0-beta: Failure to find org.deeplearning4j:deeplearning4j-examples-parent:pom:1.0.0-beta in https://repo.maven.apache.org/maven2 was cached in the local repository, resolution will not be reattempted until the update interval of central has elapsed or updates are forced and 'parent.relativePath' points at wrong local POM @ line 5, column 13

Experimental design

All of my issues pointed out have been addressed satisfactorily.

Validity of the findings

All of my issues pointed out have been addressed satisfactorily.

Additional comments

The authors have addressed all of my issues thoroughly. With the exception of one issue with the provided raw data file that is probably easy-to-fix (sorry for not pointing it out earlier), I am happy to recommend acceptance of the paper.

---

## Round 0.3 · accepted · Accept

The main issue raised in the previous round was the lack of usage instructions, which are now included, and hence the paper can be accepted in the present form.